# Measuring Topic Quality using Word Buckets

## Abstract

Measuring topic quality is essential for scoring the learned topics and their subsequent use in Information Retrieval and Text classification. To measure quality of Latent Dirichlet Allocation (LDA) based topics learned from text, we propose a novel approach based on grouping of topic words into buckets (TBuckets). A single large bucket signifies a single coherent theme, in turn indicating high topic coherence. TBuckets represents topic words using their word embeddings and employs 3 different techniques for creating buckets of words - i) clustering based, ii) using singular value decomposition (SVD) and iii) SVD with reorganization. The TBuckets approach outperforms the state-of-the-art techniques when evaluated using three publicly available datasets. Further, we demonstrate the usefulness of TBuckets for the task of weakly supervised text classification.

## 1 Introduction

Starting with the formalization of the notion of a topic as a probability distribution over words, probabilistic graphical models have been widely investigated for inferring the set of topics present in a document collection in an unsupervised manner (Blei et al., 2003). These models also infer the probability distribution over topics for documents in the collection. Since topics give a particular perspective on the structure of the document collection, topic modelling techniques have been applied on a variety of real-life document collections, such as scientific papers (Griffiths and Steyvers, 2004), (Blei, 2012) and newspapers archives (Yang et al., 2011). Topic models have also been used for improving many traditional text-mining tasks, such as document classification (Hingmire et al., 2013), document summarization (Wang et al., 2009), sentiment analysis (Lin and He, 2009), word sense disambiguation (Boyd-Graber et al., 2007), corpus visualization (Newman et al., 2010a) etc. Several variations on topic models are also being researched; e.g., correlated topic models (e.g., a document having a topic environment is likely to include topics such as UN and politics but not sports) (David M. Blei, 2007), dynamic topic models evolving over time (Blei and Lafferty, 2006), (Wang et al., 2008) and supervised topic models (Mcauliffe and Blei, 2008), (Ramage et al., 2009).

Given this growing importance of topic modelling in text mining techniques and in practical applications, it is crucial to ensure that the inferred topics are of as high quality as possible. An attractive feature of the probabilistic topic models is that the inferred topics can be easily interpreted by humans, each topic being just a bag of probabilistically selected "prominent" words in that topic's distribution. This has opened up a research area which explores using human expertise or designing automated techniques to measure the quality of topics and improve the topic modelling techniques by incorporating these measures. As an example, consider the following two topics inferred from a document collection:

{loan, foreclosure, mortgage, home, property, lender, housing, bank, homeowner, claim}

{horse, sullivan, business, secretariat, owner, get, truck, back, old, merchant}

The first topic is easily interpretable by humans whereas the second topic is much less coherent and hence less understandable.

One could evaluate a single topic or an entire set of topics ("topic model") for quality. Several different approaches have been proposed in the

literature for measuring the quality of a particular topic or that of an entire topic model: word and topic intrusions (Chang et al., 2009), analysis of the topic word probability distributions (Al-Sumait et al., 2009), average pointwise mutual information (PMI) between topic words (Newman et al., 2010b), co-document frequencies of the topic words (Mimno et al., 2011), coverage and specificities of WordNet hierarchies for words in a topic (Musat et al., 2011), distributional semantics (distances between vectors for words in a topic) (Aletras and Stevenson, 2013), among many others.

In this paper, we propose a novel approach TBuckets which groups topic words into thematic groups (which we call buckets). The intuition is that if a single large bucket is obtained from a topic, then the topic carries a single coherent theme. Under TBuckets, we explore three techniques for creating buckets of words - i) clustering based, ii) using singular value decomposition (SVD) and iii) SVD with reorganization. We evaluate our techniques by correlating their estimated coherence scores with human annotated scores and compare with state-of-the-art results reported in Roder et al. (2015). The TBuckets approach not only outperforms the state-of-the-art, but its SVD-based techniques carry merit for being completely parameter free.

This rest of the paper is organized as follows. Section 2 briefly discusses the necessary background. Sections 3 describes the TBuckets approach in detail. Section 4 gives experimental evaluation of our techniques. Section 5 discusses the relevant related work and section 6 concludes with a discussion on future work.

## 2 Background

### 2.1 Word Embeddings

Word embeddings are vector representations of words which have become popular recently. Some efficient approaches to learn these vector representations from large unlabeled corpora are by Mikolov et al. (2013) and Pennington et al. (2014). Each word is mapped to a real-valued vector in $d$ dimensions, such that vectors of semantically similar words lie close to each other. The cosine similarity of word vectors in this space is hence, a good estimation of semantic similarity between the corresponding words.

### 2.2 Singular Value Decomposition (SVD)

Through SVD, a rectangular matrix $A$ can be factorized to three components $U$, $S$ and $V$ where the matrix $U$ contains eigenvectors of $AA^T$, the matrix $V$ contains the eigenvectors of $A^T A$ and the matrix $S$ is a diagonal matrix containing the singular values of $A$, which are effectively square roots of eigenvalues of $AA^T$ and $A^T A$. Intuitively, the SVD of a rectangular matrix, allows to express the matrix as a combination of three geometrical operations - rotation through $U$, scaling through $S$ and another rotation through $V$. More clearly, for an $m \times n$ matrix $A$, where $m$ entities are represented by their $n$ features, the $U$ matrix helps to identify important dimensions among the entities and the $V$ matrix does so for important dimensions among the entities in terms of their features. In the paper, we focus particularly on the $V$ matrix allowing us to obtain eigenvectors of the matrix $A^T A$, which captures feature-feature interactions of the $m$ entities.

## 3 TBuckets: Creating buckets of topic words

The TBuckets idea is based on how we humans generally observe a topic and conclude on its coherence. Assuming a topic with words ordered (descending) by their probability of getting generated from the topic, the general procedure involves observing the topic words one by one and putting them in some form of logical groups (or *buckets*, as we call them). Starting with a fresh bucket for the first word, every new word is put in an already created bucket if the word is semantically similar or semantically associated with the words in the bucket; otherwise the word is put in a new bucket. On completion of this exercise, all topic words would be distributed in various buckets. A distribution with a single large bucket and few small buckets would signify better coherence. On the other hand, a distribution with multiple medium sized buckets would indicate lower coherence.

To see an example dry run of the above procedure, lets consider a coherent and a non-coherent topic. The topic {storm, weather, wind, temperature, rain, snow, air, high, cold, northern} is quite coherent and deals with weather and its factors. The bucket procedure for this topic executes as follows:

1. Word Seen: storm.
   Bucket-1: {storm}

2. Word Seen: `weather`.
   Bucket-1: {`storm, weather`}

3. Word Seen: `wind`.
   Bucket-1: {`storm, weather, wind`}

4. Word Seen: `temperature`.
   Bucket-1: {`storm, weather, wind, temperature`}

5. Word Seen: `rain`.
   Bucket-1: {`storm, weather, wind, temperature, rain`}.

6. Word Seen: `snow`.
   Bucket-1: {`storm, weather, wind, temperature, rain, snow`}.

7. Word Seen: `air`.
   Bucket-1: {`storm, weather, wind, temperature, rain, snow, air`}.

8. Word Seen: `high`.
   Bucket-1: {`storm, weather, wind, temperature, rain, snow, air`};
   Bucket-2: {`high`}

9. Word Seen: `cold`.
   Bucket-1: {`storm, weather, wind, temperature, rain, snow, air, cold`};
   Bucket-2: {`high`}

10. Word Seen: `northern`.
    Bucket-1: {`storm, weather, wind, temperature, rain, snow, air, cold`};
    Bucket-2: {`high`};
    Bucket-3: {`northern`}

Another topic {`karzai, afghan, miner, official, mine, assange, government, kabul, afghanistan, wikileaks`} is not coherent and deals with multiple areas like afghanistan, wikileaks and mining. The bucket procedure for this topic executes as follows:

1. Word Seen: `karzai`.
   Bucket-1: {`karzai`}

2. Word Seen: `afghan`.
   Bucket-1: {`karzai, afghan`}

3. Word Seen: `miner`.
   Bucket-1: {`karzai, afghan`};
   Bucket-2: {`miner`}

4. Word Seen: `official`.
   Bucket-1: {`storm, weather`};
   Bucket-2: {`miner`};
   Bucket-3: {`official`}

5. Word Seen: `mine`.
   Bucket-1: {`karzai, afghan`};
   Bucket-2: {`miner, mine`};
   Bucket-3: {`official`}

6. Word Seen: `assange`.
   Bucket-1: {`karzai, afghan`};
   Bucket-2: {`miner, mine`};
   Bucket-3: {`official`};
   Bucket-4: {`assange`}

7. Word Seen: `government`.
   Bucket-1: {`karzai, afghan`};
   Bucket-2: {`miner, mine`};
   Bucket-3: {`official, government`};
   Bucket-4: {`assange`}

8. Word Seen: `kabul`.
   Bucket-1: {`karzai, afghan, kabul`};
   Bucket-2: {`miner, mine`};
   Bucket-3: {`official, government`};
   Bucket-4: {`assange`}

9. Word Seen: `afghanistan`.
   Bucket-1: {`karzai, afghan, kabul, afghanistan`};
   Bucket-2: {`miner, mine`};
   Bucket-3: {`official, government`};
   Bucket-4: {`assange`}

10. Word Seen: `wikileaks`.
    Bucket-1: {`karzai, afghan, kabul, afghanistan`};
    Bucket-2: {`miner, mine`};
    Bucket-3: {`official, government`};
    Bucket-4: {`assange, wikileaks`}

It is evident from the above iterations that final distributions of topic words into buckets, reflects the coherence of a topic quite closely. Based on this idea, we devise approaches to carry out topic word distribution into buckets, automatically. We consider properties of the finally formed buckets to compute a coherence score for a topic.

The TBuckets approach powers us to do this bucketing automatically and generates a coherence score for a topic. It mainly requires two resources namely the word embeddings of topic words and the topic model (i.e. complete set of topics with words ordered according to their generation probability). These resources are not difficult to obtain as the topic model is available de facto and word embeddings of a large set of words, trained on various corpora, are now available publicly. The approach can be characterized through three techniques.

The first technique (TBuckets-Clustering) is based on a the idea of clustering which arises intuitively when we think of forming related groups among a set of items (words here). The TBuckets-Clustering technique involves representing words by their word embeddings and clustering them using agglomerative clustering. We use a maximum distance threshold as an input to the clustering for facilitating the agglomeration. Further we try with all single, average and complete linkage configurations. The technique proves to be competent involving only a single parameter.

The notion for the SVD-based technique (TBuckets-SVD) is to find the various orthogonal

dimensions or sub-themes inside a topic and attach words to those sub-themes. The technique starts by constructing a matrix of $d$ dimensional word embeddings of $n$ words of a topic leading to a $n \times d$ rectangular matrix $A$. An SVD operation on $A$ provides the orthonormal bases in $U$ and $V$ and singular values in $S$. We focus particularly on the $V$ matrix, as that records the eigenvectors of $A^T A$, which are in turn the orthogonal $d$ dimensional directions we are interested in. This is intuitive as a $d$ dimensional vector would represent some definite concept in the space of word embeddings and can become a distinct bucket identifier. Further, the $V$ matrix, provides eigenvectors ordered according to most significant singular values first. We use the first $n$ eigenvectors as bucket identifiers to attach words to. The attachment is simple - the word goes to the bucket represented by the word's most similar eigenvector.

### 3.1 Reorganization over TBuckets-SVD

Observation of buckets getting formed by TBuckets-SVD revealed that some words in spite of being similar/associated with words in the largest bucket, were being put to another bucket due to high similarity with that eigenvector. An example of such a topic is - {show, television, tv, news, network, medium, fox, cable, channel, series}. The output of TBuckets-SVD reveals the distribution - Bucket 1: {show, television, tv, network, cable, channel}; Bucket 2: {medium}; Bucket 3: {series}; Bucket 4: {news}; Bucket 5: {fox}. We can clearly observe that words news, fox and series belonging to other buckets should be a part of the largest bucket (Bucket 1). Many such similar examples were observed and hence the need for a reorganization step over TBuckets-SVD was desirable. Moreover, there were examples of topics where some words in the largest bucket were a better fit to other buckets than the current one and hence needed to be evicted for better coherence of the largest bucket. An example is a topic where TBuckets-SVD generates the largest bucket as {cocktail, glass, drink, girl}, where it is important to shift the word girl out from the bucket.

We collectively denote both the reorganization steps i.e. *into* and *from* the largest bucket, as the TBuckets-SVD-Reorg technique. The methodology for automatic reorganization is based on four important parameters for each word:

- Native Bucket Pull - Words (NBPW): Maximum semantic similarity of the word with all other words in the word's native bucket.

- Native Bucket Pull - EigenVector (NBPE): Similarity of the word with the eigenvector representing the word's native bucket.

- Largest Bucket Pull - Words (LBPW): Maximum semantic similarity of the word with all words in the largest bucket.

- Other Bucket Pull - Words (OBPW): Maximum semantic similarity of the word with all words in a bucket other than the native bucket.

Now, it is intuitive that for a word in the non-largest bucket, a condition where LBPW > NBPE, is a indication for carrying out the *into* step and for a word in the largest bucket, a condition where OBPW > max(NBPW, NBPE) is an indication for carrying out the *from* step. However, considering the *into* case, even a marginal difference (say of the order 0.01 or less) between LBPW and NBPE would lead to the shift. A similar argument holds for the *from* case. Hence, an additive threshold to the Right Hand Side (RHS) in both conditions would be necessary. We do not decide the threshold empirically, instead devise it based on the topic under consideration and hence removing possibility of any parameter tuning. The dynamic threshold depends on two values namely average word pair similarity (AWPS) among the topic words and average word eigenvector similarity (AWES). The former value is computed as an average of semantic similarity among all pairs of words in the topic. The latter value is obtained after the buckets are formed using TBuckets-SVD and the value is computed as an average of similarities between each word and its corresponding bucket's eigenvector. The threshold is computed as $AWES - \frac{AWPS}{AWES}$. The ratio in the threshold denotes flexibility among words for reorganization. A higher value of the ratio indicates better word-word similarities and lenient word-eigenvector similarities making reorganization meaningful. The subtraction from AWES manages the gap in the ranges of word pair similarities and word eigenvector similarities.

## 3.2 Computation of coherence score

For all techniques we consider the properties of words in the largest bucket to compute the coherence score. This scoring is in line with the important idea - *more topic words belonging to a common theme signifies higher coherence*. The coherence is computed based on various word and bucket properties like length of the largest bucket, function of word order in the topic, similarity to the eigenvector and most significant singular value.

Consider the largest bucket obtained from a topic, Bucket 1: $\{w_0, \cdots w_{k-1}\}$. A basic coherence score is the **size** of Bucket 1 which is $k$ in this case. We also consider the order of words in a topic which is based on their generation probability. A weight is computed for each $i^{th}$ word in a topic as follows:

$$LDA\_Order\_Weight(w_i) = \frac{1}{log(i+1)} \, if \, i > 0$$
$$= 1 \, otherwise$$

Topic coherence score based on LDA word order, denoted by **WL**, is then computed as the sum of $LDA\_Order\_Weight$ of each word in the largest bucket. For TBuckets-SVD and TBuckets-SVD-Reorg, another topic coherence score, denoted as **EVS**, can be computed using each word's similarity with its corresponding eigenvector. For reporting results for TBuckets-SVD and TBuckets-SVD-Reorg techniques, we weight the scores (size, WL and EVS) with the square of the highest singular value.

## 4 Experimental Analysis

### 4.1 Datasets

We evaluate TBuckets on 3 datasets - 20 News-Groups (20NG), New York Times (NYT) and Genomics. Each dataset consists of a set of 100 topics where each topic is represented by its 10 most probable words. Moreover, each topic is associated with a real number between 1 and 3 indicating human judgement of its coherence. Detailed description of these datasets is provided in Roder et. al (2015).

For all our experiments, we use the 300 dimensional pre-trained word embeddings provided by the GloVe framework [1]. These embeddings have

---

[1] http://nlp.stanford.edu/projects/glove/

been trained on Wikipedia and Gigaword corpora.

## 4.2 Evaluation

We use the same evaluation scheme used in (Röder et al., 2015). Each technique generates coherence scores for all the topics in a dataset. Pearson's $r$ correlation co-efficient is computed between the coherence scores based on human judgement and the coherence scores automatically generated by the technique. Higher the correlation with human scores, better is the performance of the technique at measuring coherence.

Table 1 shows the Pearson's $r$ values for our techniques - TBuckets-Clustering, TBuckets-SVD and TBuckets-SVD-Reorg. We compare their performance with the best performing correlation values as reported by Roder et al. (2015).

As observed in Table 1, our technique TBuckets-SVD-Reorg outperforms the state-of-the-art on 2 out of 3 datasets, namely NYT and 20NG. This is significant considering the fact that both SVD based techniques are completely parameter less whereas the state-of-the-art requires considerable tuning of multiple parameters. This also is a sound validation of the TBuckets idea for measuring topic coherence. TBuckets-Clustering also performs at par with the state-of-the-art on the Genomics dataset. It requires only one parameter, namely Max_Dist which is maximum allowable distance for merging in Agglomerative clustering. Figure 1 shows the variation in the performance of TBuckets-Clustering for various values of Max_Dist. Table 2 shows some examples of topics and the buckets which were created for them using the TBuckets-SVD-Reorg technique.

## 4.3 Application to Text Classification

Topic models have been used for text classification with weak supervision in several previous approaches like Hingmire et al. (2013), Hingmire and Chakraborti (2014) and Pawar et al. (2016). The core idea used in these approaches is that instead of obtaining gold labels for documents, the human annotators can provide labels for the learned topics. This reduces the labelling effort drastically as it was reported that a small number of topics (usually twice the number of class labels) was sufficient to achieve good classification performance.

As learning topics from a text corpus is based on approximate inference, different topics are generated when the process of learning topics is run

| | Setting/Scoring Metric | NYT | 20NG | Genomics |
|---|---|---|---|---|
| Roder et al. (2015) | Best performing settings | 0.806 | 0.859 | 0.773 |
| TBuckets-Clustering | Single-linkage, size | 0.779 (0.63) | 0.832 (0.7) | **0.774** (0.67) |
| | Single-linkage, WL | 0.766 (0.63) | 0.848 (0.72) | 0.751 (0.69) |
| | Average-linkage, size | 0.745 (0.81) | 0.856 (0.79) | 0.709 (0.77) |
| | Average-linkage, WL | 0.73 (0.81) | 0.863 (0.79) | 0.721 (0.86) |
| | Complete-linkage, size | 0.709 (0.91) | 0.806 (0.86) | 0.581 (0.81) |
| | Complete-linkage, WL | 0.702 (0.91) | 0.835 (0.86) | 0.567 (0.81) |
| TBuckets-SVD | size | 0.758 | 0.867 | 0.698 |
| | WL | 0.772 | 0.868 | 0.694 |
| | EVS | 0.762 | 0.854 | 0.693 |
| | WL*EVS | 0.772 | 0.858 | 0.69 |
| TBuckets-SVD-Reorg | size | 0.837 | 0.863 | 0.68 |
| | WL | **0.837** | 0.866 | 0.685 |
| | EVS | 0.828 | 0.868 | 0.687 |
| | WL*EVS | 0.833 | **0.869** | 0.691 |

size: Size of the largest bucket
WL: Word order based on generation probability from LDA
EVS: Eigenvector similarity

Table 1: Comparative performance of various techniques in terms of Pearson's $r$ correlation co-efficient

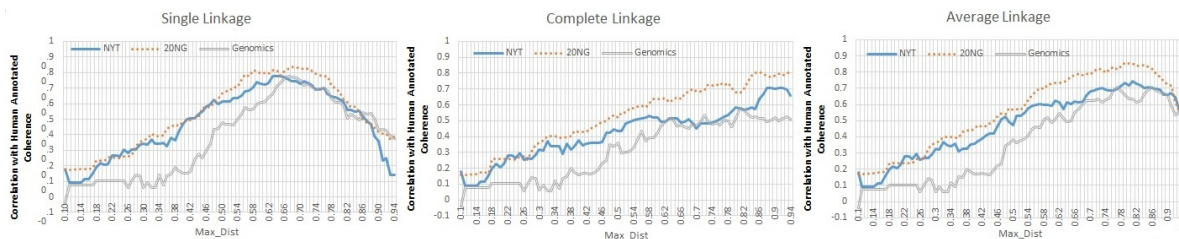

Figure 1: Effect of varying Max_Dist for TBuckets-Clustering (size)

| Dataset | Buckets | TBuckets-SVD-Reorg | Human |
|---|---|---|---|
| 20NG | Bucket 1: { gun, crime, firearm, weapon, handgun, law, criminal, control}<br>Bucket 2: { rate }<br>Bucket 3: { 000 } | 0.542 | 2.3125 |
| 20NG | Bucket 1: { convex }<br>Bucket 2: { oracle, opinion, expressed}<br>Bucket 3: { user }<br>Bucket 4: { princeton, tamu}<br>Bucket 5: { corporation }<br>Bucket 6: { phil }<br>Bucket 7: { phoenix } | 0.028 | 1.23 |
| NYT | Bucket 1: { show, television, tv, network, cable, channel, series, news, fox}<br>Bucket 2: { medium } | 0.796 | 2.82 |
| NYT | Bucket 1: { portugal, portuguese}<br>Bucket 2: { dinosaur, fossil}<br>Bucket 3: { apple }<br>Bucket 4: { ant }<br>Bucket 5: { rent, peru}<br>Bucket 6: { sherman }<br>Bucket 7: { evans } | 0.071 | 1.25 |

TBuckets-SVD-Reorg coherence scores are normalized to lie between 0 and 1
Human assigned coherence scores lie between 1 and 3

Table 2: Examples of buckets created using TBuckets-SVD-Reorg

multiple times. Hence, the process of learning topics is repeated several times and the average of classification accuracies obtained over these runs is considered as the final accuracy. Labels by human annotators have to be obtained for the set of topics generated in each of these runs. As reported in (Pawar et al., 2016) the quality of topics can be different for various runs. Low quality topics are difficult for human annotators to annotate and incorrect labels by them results in poor classification performance. To overcome this problem, we propose to evaluate quality of topics using TBuckets and present only high quality topics for human annotation. For computing quality of set of topics, we simply add the coherence scores of individual topics in the set. As all the set of topics have same number of topics, these scores are comparable.

Here, we consider 4 subsets (PC vs MAC, MEDICAL vs SPACE, POLITICS vs SCIENCE and POLITICS vs RELIGION) of 20NG dataset for the experiments involving text classification using topic labelling. We compare the performance of LPA-TD classifier used in (Pawar et al., 2016) using two different strategies for generating and labelling topics (Table 3):

**S1** (Without considering quality of topics) : Topic learning process is repeated for 5 times and human annotations were obtained. Reported classification accuracies are average of accuracies for all 5 runs.

**S2** (Considering quality of topics) : Topic learning process is repeated for 10 times and quality of each set of topics was measured using TBuckets-SVD-Reorg. But human annotations were obtained only for the *best* 5 sets of topics. Reported classification accuracies are average of accuracies for the corresponding 5 runs with best coherence scores. To emphasize usefulness of our coherence scores, we also obtain human annotations for remaining 5 sets of topics with lowest coherence scores and report average classification performance for them also.

It can be observed from Table 3 that strategy **S2** produces better accuracies than **S1** in 3 out of 4 datasets. Also, except for the MED-SPACE dataset, there is wide difference in the accuracies in S2 when the highest 5 sets of topics as per coherence are considered as against the lowest 5 sets. This can be attributed to the fact that MED-SPACE is considered to be one of the easiest for classification amongst the 20NG datasets and hence coherent topics are generated more often than not.

| Dataset | S1 | S2 | |
|---|---|---|---|
| | | 5 Highest Coh. sets | 5 Lowest Coh. sets |
| PC vs MAC | 66.98 | 68.47 | 67.96 |
| MEDICAL vs SPACE | 94.67 | 94.83 | 95.37 |
| POLITICS vs SCIENCE | 95.95 | 96.19 | 91.09 |
| POLITICS vs RELIGION | 85.61 | 85.23 | 82.85 |

Table 3: Text Classification performance (macro-F1) with and without considering quality of topics

For other datasets, especially PC-MAC which is considered one of the most difficult amongst the 20NG datasets, the large difference in accuracies (Highest 5 coherence sets vs Lowest 5 coherence sets) underlines the importance of measuring topic coherence before obtaining human annotations.

## 5 Related Work

LDA uses statistical relations between words like word co-occurrence while inferring topics and not semantic relations. Hence, topics inferred by LDA may not correlate well with human judgements even though they better optimize perplexity on held-out documents (Chang et al., 2009). (Chang et al., 2009) emphasize that quality of topics should depend on their human interpretability rather than purely statistical measures like perplexity.

Several authors (e.g. (Newman et al., 2010b; Mimno et al., 2011)) hypothesize that *coherence* of the most $N$ probable words of a topic capture its semantic interpretability and proposed measures to estimate coherence of topics. (Newman et al., 2010b) used the set of $N$ most probable words of a topic and computed its coherence ($C_{UCI}$) based on *pointwise mutual information* (PMI) between all possible word pairs of $N$ words. $C_{UCI}$ of a topic $t$ is computed as:

$$C_{UCI}(t) = \frac{2}{N(N-1)} \sum_{i=1}^{N-1} \sum_{j=i+1}^{N} PMI(w_i, w_j)$$

where,

$$PMI(w_i, w_j) = log \frac{P(w_i, w_j)}{P(w_i)P(w_j)}$$

Where, $P(w_i, w_j)$ is estimated based on the number of times words $w_i$ and $w_j$ co-occur in a sliding window of size 10 that moves over all the articles in Wikipedia. (Lau et al., 2014) propose a variant

of $C_{UCI}$ by using normalized PMI (NPMI) instead of PMI.

(Mimno et al., 2011) propose similar topic coherence measure ($C_{UMASS}$) that uses *log conditional probability* (LCP) instead of PMI and uses the same corpus on which topics are inferred, to estimate LCP rather than Wikipedia. $C_{UMASS}$ for a topic $t$ is computed as:

$$C_{UMASS}(t) = \frac{2}{N(N-1)} \sum_{i=2}^{N} \sum_{j=1}^{i-1} log \frac{P(w_i, w_j) + 1}{P(w_j)}$$

(Aletras and Stevenson, 2013) propose a topic coherence measure based on distributional similarity between the most $N$ probable words of the topic. In this approach, a topic word ($w_i$) is represented as a context vector ($\vec{v_i}$) over the words that co-occur with $w_i$ in Wikipedia in a window of $\pm 5$ words, such that $v_{i,j}$ represents PMI (or NPMI) between words $w_i$ and $w_j$. The word vectors of topic words are then used to find coherence ($C_{USheffield}$) of a topic ($t$) as follows:

$$C_{USheffield}(t) = \frac{\sum_{i=1}^{N} sim(T_C, \vec{v_i})}{N}$$

where, $T_C$ represents the centroid of the word vectors of the topic and $sim(\vec{u}, \vec{v})$ is cosine similarity between vectors $\vec{u}$ and $\vec{v}$.

(Aletras and Stevenson, 2013) observed that $C_{UCI}$ with NPMI correlates well with human judgements than $C_{UMASS}$ and $C_{UCI}$ with PMI.

(Aletras and Stevenson, 2013) also propose an alternative to $C_{USheffield}$ where they represent a topic word $w_i$ as a context vector over the space of topic words only. They observed that $C_{USheffield}$ with topic words only outperforms $C_{UCI}$ with NPMI.

Roder et al. (2015) propose a unifying framework that represents a coherence measure as a composition of parts, that can be freely combined to form a configuration space of coherence definitions. These parts can be grouped into four dimensions: 1) first dimension defines number of ways a word set can be divided into smaller pieces, 2) second dimension defines confirmation measures like PMI or NPMI to measure the agreement of a given word pair, 3) third dimension defines different ways to estimate word probabilities ($P(w_i)$ and $P(w_i, w_j)$), 4) fourth dimension defines methods to aggregate scalar values computed by the confirmation measure. This framework spans over a large number of configuration space of coherence measures, hence it becomes computationally expensive to find appropriate coherence measure for a set of topics.

# 6 Conclusion and Future Work

We proposed a novel approach TBuckets to measure quality of Latent Dirichlet Allocation (LDA) based topics, based on grouping of topic words into buckets. TBuckets uses 3 different techniques for creating buckets of words - TBuckets-Clustering which performs agglomerative clustering of words, TBuckets-SVD which uses singular value decomposition (SVD) to discover important sub-themes in topic words and lastly TBuckets-SVD-Reorg which reorganized buckets obtained from TBuckets-SVD. We evaluated our techniques on three publicly available datasets by correlating the estimated coherence scores with human annotated scores and demonstrated better performance than the state-of-the-art results. Moreover, as compared to the state-of-the-art technique which needs to tune multiple parameters, our techniques TBuckets-SVD and TBuckets-SVD-Reorg require absolutely no parameter tuning. We also highlighted the utility of TBuckets for the task of weakly supervised text classification.

In future, we plan to devise better ways to compute word similarities which would be more suitable for specific domains like Genomics. Furthermore, we plan to devise a unified framework to evaluate and employ various scoring metrics. Also, we wish to explore usefulness of our techniques for applications other than text classification.

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
