# Peer review of "Measuring Topic Quality using Word Buckets"

_CoNLL 2016 — decision unknown_

[Official Review · Reviewer 1 · rating 2 · confidence 4]
soundness 3 · originality 3 · clarity 4 · impact 3 · substance 3 · appropriateness 5 · meaningful comparison 5 · replicability 4 · presentation format Poster

This paper proposes a method for evaluating topic quality based on using word
embeddings to calculate similarity (either directly or indirectly via matrix
factorisation), achieving impressive results over standard datasets.

The proposed method represents a natural but important next step in the
evolutionary path of research on topic evaluation. The thing that troubled me
most with the results was that, while you achieve state-of-the-art results for
all three datasets, there are large inconsistencies in which methods perform
and which methods perform less well (below the state of the art). In practice,
none of the proposed methods consistently beats the state of the art, and the
SVD-based methods perform notably badly over the genomics dataset. For someone
who wants to take your method off the shelf and use it over any arbitrary
dataset, this is a considerable worry. I suspect that the lower results for
SVD over genomics relate to the proportion of OOV terms (see comment below),
and that it may be possible to automatically predict which method will perform
best based on vocab match with GloVe etc., but there is no such discussion in
the paper.

Other issues:

- the proposed method has strong similarities with methods proposed in the
  lexical chaining literature, which I would encourage the authors to read up
  on and include in any future version of the paper

- you emphasis that your method has no parameters, but the word embedding
  methods have a large number of parameters, which are implicit in your
  method. Not a huge deal, but worth acknowledging

- how does your method deal with OOV terms, e.g. in the genomics dataset
  (i.e. terms not present in the pretrained GloVe embeddings)? Are they simply
  ignored? What impact does this have on the method?

Low-level issues:

- in your description of word embeddings in Section 2.1, you implicitly assume
  that the length of the vector is unimportant (in saying that cosine
  similarity can be used to measure the similarity between two vectors). If
  the vectors are unit length, this is unproblematic, but word2vec actually
  doesn't return unit-length vectors (the pre-trained vectors have been
  normalised post hoc, and if you run word2vec yourself, the vector length is
  certainly not uniform). A small detail, but important.

- the graphs in Figure 1 are too small to be readable

[Official Review · Reviewer 2 · rating 3 · confidence 3]
soundness 4 · originality 3 · clarity 4 · impact 3 · substance 3 · appropriateness 5 · meaningful comparison 4 · replicability 4 · presentation format Oral Presentation

This paper proposes a new method for the evaluation of topic models that
partitions the top n words of each topic into clusters or "buckets" based on
cosine similarity of their associated word embeddings. In the simplest setup,
the words are considered one by one, and each is either put into an existing
"bucket" â if its cosine similarity to the other words in the bucket is below
a certain threshold â or a new bucket is created for the word. Two more
complicated methods based on eigenvectors and reorganisation are also
suggested. The method is evaluated on three standard data sets and in a  weakly
supervised text classification setting. It outperforms or is en par with the
state of the art (RÃ¶der et al., 2015).

The basic idea behind the paper is rather simple and has a certain ad
hoc-flavour. The authors do not offer any new explanations for why topic
quality should be measurable in terms of wordâword similarity. It is not
obvious to me why this should be so, given that topics and word embeddings are
defined with respect to two rather different notions of context (document vs.
sequential context). At the same time, the proposed method seems to work quite
well. (I would like to see some significance tests for Table 1 though.)

Overall the paper is clearly written, even though there are some language
issues. Also, I found the description of the techniques in Section 3 a bit hard
to follow; I believe that this is mostly due to the authors using passive voice
("the threshold is computed as") in places were they were actually making a
design choice. I find that the authors should try to explain the different
methods more clearly, with one subsection per method. There seems to be some
space for that: The authors did not completely fill the 8 pages of content, and
they could easily downsize the rather uninformative "trace" of the method on
page 3.

One question that I had was how sensitive the proposed technique was to
different word embeddings. For example, how would the scores be if the authors
had used word2vec instead of GloVe?